

# Evaluation of thyroid antibodies and benign disease prevalence among young adults exposed to $^{131}$I more than 25 years after the accident at the Chernobyl Nuclear Power Plant

Yuko Kimura[1], Naomi Hayashida[2], Jumpei Takahashi[3], Ruslan Rafalsky[4], Alexsey Saiko[4], Alexander Gutevich[4], Sergiy Chorniy[4], Takashi Kudo[5] and Noboru Takamura[1]

[1] Department of Global Health, Medicine and Welfare, Atomic Bomb Disease Institute, Nagasaki University, Nagasaki, Japan
[2] Division of Strategic Collaborative Research, Atomic Bomb Disease Institute, Nagasaki University, Nagasaki, Japan
[3] Center for International Collaborative Researches, Nagasaki University, Nagasaki, Japan
[4] Zhitomir Inter-Area Medical Diagnostic Center, Korosten, Ukraine
[5] Department of Radioisotope Medicine, Atomic Bomb Disease Institute, Nagasaki University, Nagasaki, Japan

Corresponding author
Noboru Takamura,
takamura@nagasaki-u.ac.jp

## ABSTRACT

**Background.** The Chernobyl Nuclear Power Plant (CNPP) accident exposed a large number of inhabitants to internal $^{131}$I radiation. The associations between internal $^{131}$I exposure and thyroid autoimmunity and benign thyroid diseases remain controversial in the population living in the contaminated area around the CNNP. In this study, we evaluate the association of $^{131}$I with benign thyroid diseases. **Methods.** We compared the prevalence of Anti-Thyroid Autoantibodies (ATAs), thyroid function, and prevalence of thyroid ultrasound finding outcomes in 300 residents of the contaminated area of Ukraine who were 0–5 years of age at the time of the CNPP accident (group 1) and 300 sex-matched residents who were born after the accident (group 2). **Results.** We did not find any differences of the prevalence of Antithyroglobulin Antibodies (TGAb) positive, Antithyroid Peroxidase Antibodies (TPOAb) positive, and TGAb and/or TPOAb positive between the study groups. (11.7% vs 10.3%; $p = 0.602$, 17.3% vs 13.0%; $p = 0.136$, 21.0% vs 17.3%; $p = 0.254$, respectively); after adjusting for age and sex, the prevalence was not associated with the $^{131}$I exposure status in the study groups. The prevalence of subclinical and overt hypothyroidism cases was not significantly different ($p = 0.093$ and $p = 0.320$) in the two groups, nor was the prevalence of goiter ($p = 0.482$). On the other hand, the prevalence of nodules was significantly higher in group 1 ($p = 0.003$), though not significantly so after adjustment for age and sex. **Discussion.** Working 26–27 years after the CNNP accident, we found no increased prevalence of ATAs or benign thyroid diseases in young adults exposed to $^{131}$I fallout during early childhood in the contaminated area of Ukraine. Long-term follow-up is needed to clarify the effects of radiation exposure on autoimmunity reaction in the thyroid.

## INTRODUCTION

The Chernobyl Nuclear Power Plant (CNPP) accident on 26[th] April 1986 was the worst nuclear disaster in history. The accident caused the release of a large amount of radionuclides into the environment (*United Nations Scientific Committee on the Effects of Atomic Radiation, 2011*). The fallout, which contained both short-lived radionuclides consisting mostly of iodine-131 ($^{131}$I) and long-lived radionuclides made up largely of cesium-137 ($^{137}$Cs) (*Christodouleas et al., 2011*), exposed a great number of inhabitants living in contaminated areas now in Ukraine, Belarus, and the Russian Federation to internal radiation of the thyroid gland through the ingestion of contaminated foods containing $^{131}$I (*United Nations Scientific Committee on the Effects of Atomic Radiation, 2011*). Several years after the accident, the incidence of childhood thyroid cancer had dramatically increased in the population living in those areas and was strongly related to this $^{131}$I exposure (*Cardis et al., 2005*; *Demidchik, Saenko & Yamashita, 2007*; *Zablotska et al., 2011*). Between 1991 and 2005, 5,127 cases of thyroid cancer were reported in residents exposed as children under 14 years old in 1986 (6,848 cases in those under 18), according to the UNSCEAR report (*United Nations Scientific Committee on the Effects of Atomic Radiation, 2011*).

On the other hand, several studies have been conducted regarding the relationship between radiation exposure and thyroid autoimmunity and benign thyroid diseases, but the results remain inconsistent (*Ito et al., 1995*; *Vykhovanets et al., 1997*; *Pacini et al., 1998*; *Vermiglio et al., 1999*). Screening studies targeting individuals under 18 years old at the time of the accident showed a significant association between thyroid $^{131}$I dose and the prevalence of subclinical hypothyroidism (*Ostroumova et al., 2009*; *Ostroumova et al., 2013*), but not with Autoimmune Thyroiditis (AIT) (*Tronko et al., 2006*; *Ostroumova et al., 2009*; *Ostroumova et al., 2013*). *Agate et al. (2008)* also reported a higher prevalence of thyroid dysfunction and TPOAb without autoimmune thyroiditis in children living in the contaminated region 13–15 years after the accident, compared to non-contaminated Belarusian settlements (*Agate et al., 2008*). Almost thirty years has passed since the Chernobyl accident. The residents who lived through the accident in childhood are now adults and the situation might have evolved. The purpose of this study was to evaluate the association of internal radiation exposure to $^{131}$I and the prevalence of anti-thyroid autoantibodies (ATAs) and benign thyroid disease among young adults living in the contaminated area of Ukraine, more than 25 years after the accident.

## MATERIAL AND METHODS

This study was conducted primarily at the Korosten Inter-Area Medical Diagnostic Center ("the center"), Korosten, Zhitomir region, Ukraine. This area is located 120 km southwest of the CNPP and was heavily affected by the accident. The estimated average $^{131}$I thyroid doses in children and adolescents living in this region at the time

of the accident was 0.15–0.65 Gy (*United Nations Scientific Committee on the Effects of Atomic Radiation, 2011*).

For study participants, we recruited two groups of young adults who visited the center between June 2012 and January 2014 for their annual health screening. The first group consisted of young adults (n = 300) born between 1st January 1981 and 26th April 1986 (age 0–5 years on 26th April 1986) who lived in the region at the time of the accident and were not evacuated (group 1). This group was considered to have experienced internal $^{131}$I radiation exposure. The second group consisted of sex-matched young adults (n = 300) born between April 1987 and December 1991 (more than one year after the accident) and residing in the Zhitomir region after the accident (group 2). This group was considered to have not experienced internal $^{131}$I radiation exposure; since the half-life of $^{131}$I is about eight days, the $^{131}$I had decayed within a few months after the accident and any effect of $^{131}$I should have been extremely low when they were born. Young adults who had a history of thyroid cancer or had undergone thyroid lobectomy were excluded. The study protocol was approved by the center's Institutional Review Board (No. 002) and the ethical committee of Nagasaki University Graduate School of Biomedical Sciences (No. 12122865). Prior to the study, written informed consent was obtained from all participants.

Blood samples were collected for measurement of Free Triiodothyronine (FT3), Free Thyroxin (FT4), Thyroid-Stimulating Hormone (TSH), Antithyroglobulin Antibodies (TGAb), and Antithyroid Peroxidase Antibodies (TPOAb). Serum $FT_3$, $FT_4$, TSH, TGAb, and TPOAb levels were measured using a Stat Fax® 303 Plus Enzyme-Linked Immunosorbent Assay (ELISA) (Awareness Technology, Inc., Palm City, FL, USA). The laboratory reference range for FT3, FT4, and TSH were 1.4–4.2 ng/ml, 0.8–2 ng/ml, and 0.3–6.2 µIU/ml, respectively. TGAb values <8 IU/ml and TPOAb values <20 IU/ml were considered negative. We defined overt hypothiroidism as freeT4 <0.8 ng/dl and TSH >6.2 µIU/ml and subclinical hypothyroidism as freeT4 ≧0.8 ng/dl and TSH >6.2 µIU/ml.

Ultrasonography of the thyroid gland was performed in both groups using 8.5 MHz Nemio XG SSA-580A probes (Toshiba, Tokyo, Japan). Presence of nodules, cysts, and echostructure were recorded. The thyroid volume was calculated based on the formula, length × width × depth × 0.479 described by *Brunn et al. (1981)*. Finally, we defined goiter as thyroid volume larger than 25 ml in men and 18 ml in women.

The $^{137}$Cs body burdens in both groups were measured using a whole-body counter (γ-spectrometer, model 101, equipped with a collimator; Aloka Co., Ltd., Tokyo, Japan). The detectable $^{137}$Cs body burden was 270 Bq/body. Participants who did not have any $^{137}$Cs exposure were defined as be "0 Bq."

Data are expressed as means plus Standard Deviations (SDs) and medians. Differences in age, FT3 and FT4 concentrations, and thyroid volumes between groups were evaluated using t-tests. TSH concentration was distributed in a skewed manner, so logarithmic transformation was performed for the analysis. FT3 and FT4 concentrations adjusted for age and thyroid volumes adjusted for body weight were compared by analysis of covariance between the groups. The difference between groups for $^{137}$Cs body burden was evaluated by a Mann-Whitney's U-test. Frequencies of
**Table 1 Characteristic and thyroid outcomes of study groups.**

|  | Group 1 (n = 300) | Group 2 (n = 300) | *p* value | Adjusted *p* value |
|---|---|---|---|---|
| Age (at the examination) | 28.3 ± 1.4 | 23.0 ± 1.4 | <0.001 | – |
| Female, n (%) | 237 (79) | 237 (79) | – | – |
| Free T3, pg/ml | 2.66 ± 1.18 | 2.75 ± 0.99 | 0.333 | 0.016* |
| Free T4, ng/dl | 1.33 ± 0.39 | 1.39 ± 0.30 | 0.014* | 0.074 |
| TSH, μIU/ml | 1.08 (0.76–1.75) | 1.22 (0.81–1.77) | 0.183 | – |
| Log(TSH) | 0.02 ± 0.44 | 0.04 ± 0.40 | 0.308 | 0.602 |
| Overt hypothyroidism, n | 3 | 1 | 0.320 | – |
| Subclinical hypothyroidism, n | 7 | 2 | 0.093 | – |
| Thyroid volume, ml | 15.93 ± 7.00 | 15.74 ± 5.29 | 0.718 | 0.204 |
| $^{137}$Cs body burden, Bq/kg | 0 (0–49.65) | 0 (0–159.34) | 0.261 | – |

**Notes:**

Age, FreeT3, FreeT4, log TSH and thyroid volume are shown as mean ± SD. TSH is shown as median(IQR) and $^{137}$Cs body burden is shown as median (minimum-maximum). FreeT3, FreeT4 concentrations and log-TSH were adjusted for age and thyroid volume was adjusted for body weight between the groups were compared by analysis of covariance.

*$p < 0.05$.

positive TGAb, TPOAb, thyroid nodules, and goiter were evaluated using the $\chi^2$-test. Frequencies of hypothyroidism were evaluated using Fisher's exact test. We used logistic regression analysis to assess the association between the prevalence of ATAs, goiter, and nodules and $^{131}$I exposure, age, and sex. All statistical analyses were performed using SPSS software, v.22 for Mac (SPSS Japan, Tokyo, Japan). The *p* values of less than 0.05 were considered statistically significant.

## RESULTS

The characteristics of participants, functional thyroid outcomes and $^{137}$Cs body burdens are summarized in Table 1. The mean age of groups 1 and 2 was 28.3 and 23.0 years old at examination, respectively. Each group included 237 females (79%). The median $^{137}$Cs body burden was below the detection limit in both groups, and no significant difference was noted between the groups ($p = 0.261$). Most participants had FT3 and FT4 concentrations within the normal range, but FT4 concentrations were significantly higher in group 2 ($p = 0.014$) and FT3 concentrations adjusted for age were significantly higher in group 2 ($p = 0.016$). Median TSH concentrations in groups 1 and 2 were 1.08 μIU/ml and 1.22 μIU/ml, respectively, but this difference was not statistically significant ($p = 0.183$). A few subclinical or overt hypothyroidism cases were observed, all of them were ATA-positive. The prevalence of subclinical hypothyroidism was not significantly different between the groups ($p = 0.093$). Only three and one overt hypothyroidism cases were observed in groups 1 and 2, respectively; the prevalence was also not significantly different ($p = 0.320$).

The prevalence of TGAb positive or TPOAb positive was not significantly higher in group 1 than in group 2 (TGAb positive: 11.7% vs 10.3%; $p = 0.602$, TPOAb positive: 17.3% vs 13.0%; $p = 0.136$, respectively, Fig. 1). The prevalence of TGAb and/or TPOAb positive was also not significantly different (21.0% vs 17.3%; $p = 0.254$). Logistic regression analysis adjusted for age and sex showed that only female gender was

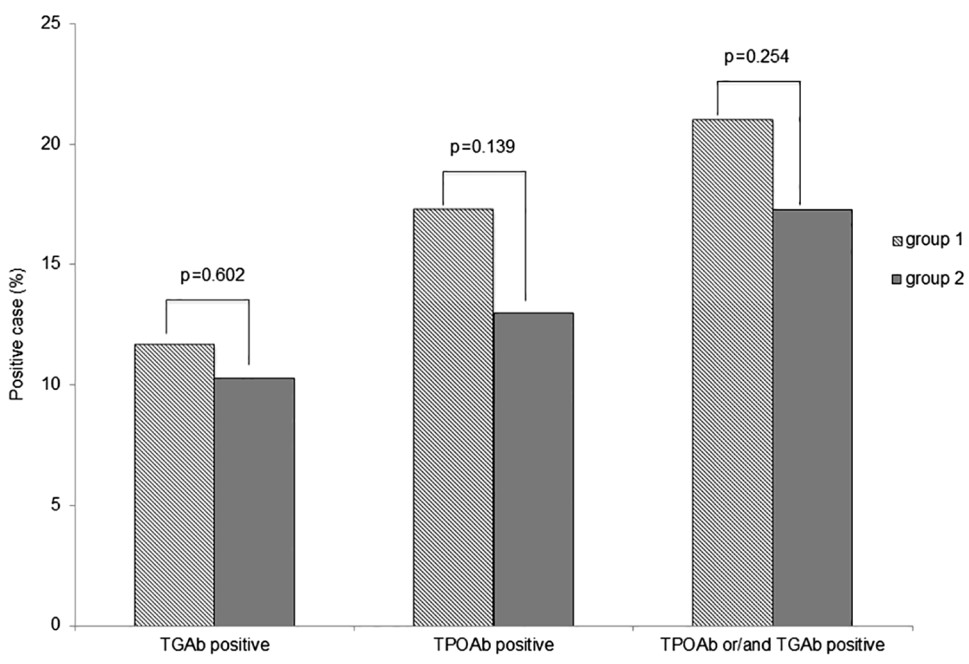

**Figure 1 Prevalence of thyroid ATAs in groups 1 and 2.** The prevalence of TGAb, TPOAb and TGAb and/or TPOAb was slightly but not significantly higher in group 1 compared with group 2.

**Table 2 Logistic regression analysis of TGAb and TPOAb with examined variables: $^{131}$I exposure, sex and age at examination.**

| Values | TGAb | | TPOAb | | TGAb and/or TPOAb | |
|---|---|---|---|---|---|---|
| | OR (95% CI) | *p* value | OR (95% CI) | *p* value | OR (95% CI) | *p* value |
| $^{131}$I exposure (group 1/group 2) | 2.029 (0.664–6.193) | 0.214 | 2.529 (0.953–6.711) | 0.062 | 1.943 (0.797–4.738) | 0.144 |
| Sex (female/male) | 9.762 (2.355–40.472) | 0.002* | 2.749 (1.338–5.648) | 0.006* | 3.775 (1.852–7.698) | <0.001* |
| Age at examination | 0.889 (0.747–1082) | 0.260 | 0.896 (0.762–1.054) | 0.184 | 1.943 (0.797–4.738) | 0.144 |

**Note:**
*$p < 0.05$.

significantly correlated with TGAb, TPOAb, and TGAb and/or TPOAb positive prevalence (Table 2).

The mean thyroid volume was 15.93 ml for group 1 and 15.74 ml for group 2, with no significant difference of prevalence of goiter in the two groups (25.3% vs 26.7%; $p = 0.780$, Fig. 2). The prevalence of thyroid nodules was significantly higher in group 1 (16.3% vs 8.3%; $p = 0.003$); however, logistic regression analysis adjusted for age and sex showed that $^{131}$I exposure status was not correlated with nodule prevalence (OR = 0.782, 95% CI 0.263–2.320, $p = 0.657$), while age and female gender were significantly correlated with nodule prevalence (Table 3).

## DISCUSSION

In this study, we evaluated the association between internal $^{131}$I exposure and the prevalence of ATAs, thyroid function, and thyroid ultrasonography outcomes among young adults living in the contaminated area of Ukraine.

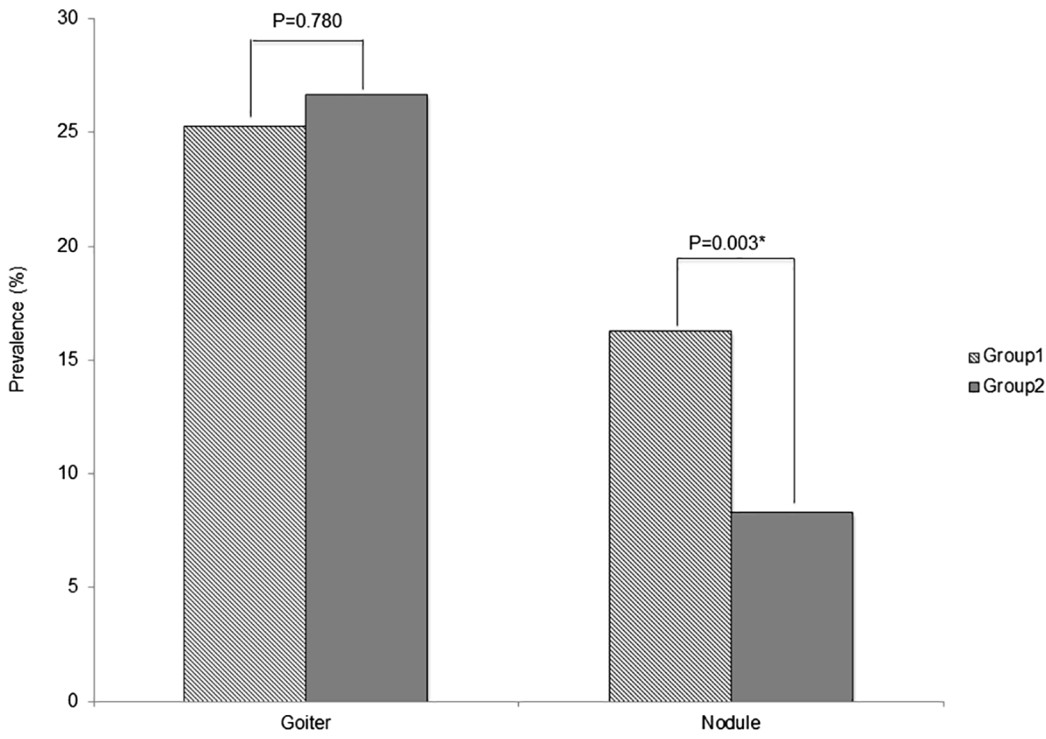

**Figure 2 Prevalence of thyroid ultrasound findings outcomes.** No significant difference of diffuse goiter prevalence was observed in both groups (19.7% vs 22.0%; $p = 0.482$). On the other hand, the prevalence of nodules was significantly higher in group 1 (16.3% vs 8.3%; $p = 0.003$). However Logistic regression analysis adjusted by age and sex showed that age and female gender were correlated with goiter and nodule prevalence.

**Table 3 Logistic regression analysis of thyroid nodule and goiter with examined variables: [131]I exposure, sex and age at examination.**

|  | Nodule | | Goiter | |
| --- | --- | --- | --- | --- |
| Values | OR (95% CI) | *p* value | OR (95% CI) | *p* value |
| [131]I exposure (group 1/group 2) | 0.782 (0.263–2.320) | 0.657 | 2.085 (0.937–4.638) | 0.072 |
| Sex (female/male) | 2.827 (1.257–6.357) | 0.012* | 2.323 (1.369–3.941) | 0.002* |
| Age at examination | 1.212 (1.011–1.452) | 0.038* | 0.860 (0.752–0.982) | 0.026* |

Note:
*$p < 0.05$.

The positive impact of this study was to evaluate the relationship between the experience of internal [131]I exposure in childhood and the occurrence of thyroid diseases more than 25 years after the accident. Additionally, by recruiting study participants who all lived in the same area, we were able to compare two groups that had similar internal [137]Cs exposure levels and iodine status, which could have confounding effects on thyroid outcomes.

Several ecological studies conducted approximately 10 years after the accident have reported that prevalence of positive TPOAb and TGAb was significantly higher in children living in areas around the CNPP with greater contamination (*Vykhovanets et al., 1997*;

*Pacini et al., 1998*; *Vermiglio et al., 1999*). On the other hand, a screening study conducted among 160,000 children aged 0–10 years at the time of the accident and living around the CNPP between 1991 and 1996 (5–10 years after the accident) within the framework of the Chernobyl Sasakawa Health and Medical Cooperation Project reported no significant relationship between the prevalence of TGAb and radiation exposure dose estimated by $^{137}$Cs body burden or with soil $^{131}$Cs contamination levels at sites where study participants were living (*Saiko et al., 1997*).

*Agate et al. (2008)* recently evaluated the prevalence of positive TPOAb, TGAb and TSH concentrations in residents living in contaminated and non-contaminated areas of Belarus, Ukraine, and the Russian Federation 13–15 years after the accident, finding that the prevalence of positive TPOAb was significantly higher in adolescents exposed to radioactive fallout in Belarus, though not in Ukraine and the Russian Federation (*Agate et al., 2008*). In our study that conducted more than 25 years after the accident, we also did not find any significant difference in ATA prevalence between young adults in Ukraine exposed to radioiodine in childhood and young adults without exposure.

We showed that FT3 concentration adjusted for age was significantly higher in group 2, but did not find any significant increase in TSH concentration or in the prevalence of antibody-positive hypothyroidism in group 1. There are large cohort studies that reported the relationship between radiation exposure due to $^{131}$I by the accident at CNNP and hypothyroidism. *Ostroumova et al. (2009)* and *Ostroumova et al. (2013)* reported screening studies conducted in Ukraine and Belarus 10–17 years after the accident in individuals under the age of 18 years at the time of the CNPP accident, showing a significant association between $^{131}$I thyroid dose and the prevalence of subclinical hypothyroidism. However, excess odds ratio was higher in individuals with TPOAb $\leq$60 U/ml than in those with TPOAb >60 U/ml in both studies (*Ostroumova et al., 2009*; *Ostroumova et al., 2013*). In same study of Ukraine, *Tronko et al. (2006)* did not find any association between $^{131}$I thyroid dose and the prevalence of autoimmune thyroiditis, but the prevalence of elevated ATPO demonstrated a modest, significant association with $^{131}$I that was well described by several concave models (*Tronko et al., 2006*). As ecological study, *Agate et al. (2008)* reported that ATA prevalence in exposed Belarusian adolescents 13–15 years after the accident was much lower than the rate in their previous study of 6–8 years after the accident, suggesting a transient autoimmune reaction that did not trigger autoimmune disease and had no effect on thyroid function (*Pacini et al., 1998*; *Agate et al., 2008*). In our study, the sample size was small so that a careful evaluation is needed, but we did not find a significant increase of antibody-positive/negative hypothyroidism in the group exposed to $^{131}$I in their childhood, 25 years after the accident in Ukraine. The thyroid immune reaction to radiation exposure remains obscure. In atomic bomb survivors exposed to acute gamma radiation, a convex dose-response relation with antibody-positive hypothyroidism was observed (*Nagataki et al., 1994*), but was not found in a more recent study (*Imaizumi et al., 2006*). Hence, the autoimmune reaction to radiation exposure in thyroid may change over time, so that further long-period observation is needed to evaluate fully the

dynamic of any relationship between radiation exposure and hypothyroidism and autoimmunity reaction in the thyroid.

Several studies on the role of radiation in the development of thyroid nodules among the cohorts of medically irradiated patients, residents exposed to fallout from nuclear testing, or atomic bomb survivors have been published (*Ron & Brenner, 2010*). *Imaizumi et al. (2006)* reported that among atomic bomb survivors, a significant linear dose-response relationship was observed for the prevalence of all solid nodules, not only malignant tumors, but also benign nodules and cysts (*Imaizumi et al., 2006*). On the other hand, in the Hanford nuclear test site study, there was no evidence that the incidence of benign thyroid nodules increased with dose ($p = 0.68$), with an estimated slope of $-0.8\%$/Gy (95% CI, $<-2.2\%$/Gy to $4.1\%$/Gy) (*Davis et al., 2004*). In the present study, the prevalence of thyroid nodules was significantly higher in group 1, but this difference disappeared after adjustment for age and sex. Age and sex are important factors influencing the development of benign thyroid nodules, and it has been reported that, even among people in their twenties, the prevalence of thyroid nodule is higher in female and older age cohorts (*Reiners et al., 2004*). In the Chernobyl Sasakawa Health and Medical Cooperation Project, the prevalence of thyroid nodules was also age-dependent and higher in girls than in boys, a fact that explains their higher incidence in older age groups (*Panasyuk et al., 1997*). In the present study, as the number of females was the same in both groups, the increase of thyroid nodule prevalence in group 1 might be due to aging rather than $^{131}$I exposure in their childhood.

In the present study, we have shown that female gender was significantly correlated with ATA prevalence and that age and female gender were significantly correlated with goiter. It is well known that aging and female gender are risk factors of high prevalence of ATAs and goiter (*Hollowell et al., 2002*; *Reiners et al., 2004*; *Hoogendoorn et al., 2006*).

This study has several limitations. The number of subjects was relatively small since it was conducted only among the population residing in Ukraine's Zhitomir region and visiting the center for an annual health screening. As a reference, in the framework of the Chernobyl Sasakawa Health and Medical Cooperation Project, around 18,000 children aged 0–5 years old at the time of the accident were screened at the center from 1991–1996 (*Yamashita & Shibata, 1997*). Therefore, we screened around 1.6% of the number of patients who accessed the center in 1991–1996. Additionally, there might be a bias in the selection of study participants, as this study's subjects were those residents who chose to visit the center and undergo health screening. Therefore, we need to evaluate carefully whether our data can be generalized to all eligible residents. We also could not evaluate individual thyroid doses because the data of thyroid dose was not available. However, The estimated average $^{131}$I thyroid doses in children living in this region at the time of the accident was reported as 0.15–0.65 Gy (*United Nations Scientific Committee on the Effects of Atomic Radiation, 2011*). And there is a ultrasound screening study of Belarusian school children living within 150 km of the CNNP showed high rates of thyroid cancer in those who were bone before the accident (exposed group; n = 31/9720, 0.32%), while no thyroid cancer was seen in those who were bone after the

accident (unexposed group; n = 0/2409) (*Shibata et al., 2001*). This result suggested that the fallout of short-lived radionuclides from the CNPP accident affected to children who were bone before the accident but did not affect to children who were bone after the accident. Therefore, we considered that it might be possible to evaluate the relationship between the experience of exposure to [131]I in childhood and prevalence of ATA and thyroid benign diseases even though the individual thyroid dose were not available.

In conclusion, we showed no increased prevalence of ATAs and benign thyroid diseases more than 25 years after the CNPP accident in young adults exposed to [131]I fallout during their childhood. Long-term follow-up in the population living around the CNPP is needed to clarify the effects of radiation exposure on the autoimmunity reaction in the thyroid.

### Funding
This study was financially supported by the Uehara Memorial Foundation and by a Grant-in-Aid from the Japan Society for the Promotion of Science (No. 26305025). The funders had no role in study design, data collection and analysis, decision to publish, or preparation of the manuscript.

### Grant Disclosures
The following grant information was disclosed by the authors:
The Uehara Memorial Foundation and by a Grant-in-Aid from the Japan Society for the Promotion of Science: 26305025.

### Competing Interests
The authors declare that they have no competing interests.

### Author Contributions
- Yuko Kimura conceived and designed the experiments, performed the experiments, analyzed the data, contributed reagents/materials/analysis tools, wrote the paper, prepared figures and/or tables.
- Naomi Hayashida contributed reagents/materials/analysis tools, reviewed drafts of the paper.
- Jumpei Takahashi contributed reagents/materials/analysis tools.
- Ruslan Rafalsky performed the experiments.
- Alexsey Saiko performed the experiments.
- Alexander Gutevich performed the experiments, reviewed drafts of the paper.
- Sergiy Chorniy performed the experiments.
- Takashi Kudo conceived and designed the experiments, analyzed the data, wrote the paper, reviewed drafts of the paper.
- Noboru Takamura conceived and designed the experiments, analyzed the data, reviewed drafts of the paper.

## Human Ethics

The following information was supplied relating to ethical approvals (i.e., approving body and any reference numbers):

The study protocol was approved by the Institutional Review Board of Korosten Inter-Area Medical Diagnostic Center (No. 002) and the ethical committee of Nagasaki University Graduate School of Biomedical Sciences (No. 12122865).

## Data Deposition

Raw data can be found in the Supplemental Information.

## Supplemental Information

Supplemental information for this article can be found online at http://dx.doi.org/10.7717/peerj.1774#supplemental-information.

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
