# Peer review of "Evaluation of thyroid antibodies and benign disease prevalence among young adults exposed to 131I more than 25 years after the accident at the Chernobyl Nuclear Power Plant"

_PeerJ, doi:10.7717/peerj.1774_

## Round 0.1 · original submission · Major Revisions

The aim of this study was to determine the association between childhood exposure to I-131 and benign thyroid disease in later life.

Two reviewers have given comments that are given below. Some serious concerns about the methodology are outlined, and there are several question on methodology and conclusions that need to be answered.

Finally, please check grammar and spelling. It is advised to have the paper read and edited by an English-native speaker.

Reviewer 1 ·

Basic reporting

The paper presents the results of a cross-sectional study on prevalence of anti-thyroid antibodies, hypothyroidism, thyroid nodules and goiter in 300 residents of Zhytomir region who were <5 years of age at the time of the Chernobyl accident. Authors compared frequencies of selected thyroid outcomes in this group with potential exposure to Iodine-131 (I-131) with respective frequencies in 300 sex-matched residents of Zhytomir region who were born after the accident.

Taking into account how the study main goal was formulated and how the study was designed and carried out, the study couldn’t provide a robust answer because there is no information on I-131 exposure levels.

Experimental design

There are several serious remarks about the study methodology. Studying an association between possible I-131 exposure and thyroid function and autoimmunity, authors have no information about possible levels of I-131 exposure in the group 1 (exposed). They assumed that those people were exposed to I-131 because they are residents of Zhytomir region. Did the authors made an attempt to verify place and time of residence in the period of substantial releases of I-131 crucial for thyroid dose formation in the subjects in group 1? The purpose to measure and compare Cesium-137 content in two study groups is unclear.
Another major concern is relatively small size of the study groups. It is unclear how the study sample of 300 people who were < 5 years old at the time of the accident and presumably exposed to I-131 (group 1), was drawn, out of how many eligible subjects?

Validity of the findings

The importance of the study findings and their contribution to our knowledge about radioiodine and thyroid function are disputable.

The results in Table 3 and Fig.2 look contradictory. An OR estimate for I-131 exposure status and thyroid nodules presence is non-significantly below 1, while the frequency of thyroid nodules is significantly higher in exposed group as compared to controls. Please, explain why it is so.

Additional comments

The manuscript needs a substantial revision and proof-reading by an English-native speaker before it could be recommended for publication.

Annotated reviews are not available for download in order to protect the identity of reviewers who chose to remain anonymous.

·

Basic reporting

The aim of this study is to determine the association between childhood exposure to I-131 and benign thyroid disease in later life. The address this aim the authors used a retospective cohort design comparing an exposed and unexposed group. Inherent to this type of study, the results can be subject to inclusion bias and confounding.

Experimental design

It would be helpful if the authors could further substantiate the statement that the control group had not been exposed to irradiation.

It is not immediately clear how the groups were recruited and selected, in particular the control group. Please provide a detailed description of the recruitment proces.

Could the authors explain why the male/female ratio is 20/80 and why this ratio is exactly the same in both groups, whereas the age is different?

The threshold for positive thyroid antibodies is unusually low, which explains the relatively large number of subjects with thyroid autoimmunity. What is the basis for this threshold?

Addition of a power calculation will aid the reader in understanding the difference between groups that could have been detected (or missed).

Validity of the findings

In the legend of figure 2 the authors state: However Logistic regression analysis adjusted by age and sex showed that age and female gender were correlated with goiter and nodule prevalence.
Was the difference in nodule prevalence between groups still significant after correction for age and sex?

Additional comments

Please check grammar and spelling.

---

## Round 0.2 · Major Revisions

The authors revised the manuscript and anwered to the comments of the reviewers.

Reviewer 1 has extensively responded to this revision and points out several separate small points that need attention but also some more extensive comments on the validity of the studies. Especially the remark on the novelty of the study needs attention in a revised version. please respond to these comment point to point and make adjustments were appropriate.

The comments of reviewer 2 were answered, however one comment still stands and was not answered in the point-to-point response which is on the power analysis and whether the authors have power enough in their study set up to support the conclusion. It is important to also comment on this and make appropriate revisions.

Reviewer 1 ·

Basic reporting

Abstract:
Line 32: please change for “benign thyroid diseases”, plural
Line 33: Please correct for anti-thyroid, “i” is missing
Line 37: abbreviations of TGA and TPOAb are not spelled out.
Line 40: change for “the prevalence was not associated with the I-131 exposure status in the study groups”
Line 41: it is confusion to see to two p-values presented for hypothyroidism prevalence. It needs explanation that it is for overt and subclinical hypothyroidism prevalences.
Line 74: Reference has different format. Please, correct.
Materials and Methods:
Line 87-93: para present materials and methods and even results of absolutely different study by Shibata et al. Please remove this para and, if necessary, mention the study by Shibata et al. in the Discussion.
Line 121: please, delete “was”, it should read as “as a thyroid volume larger than 25 ml…”
Line 135: please change for “nodules”, plural.
Results
Line 149: please replace for “them” not “whom”
Line 150: please delete “case”, it should read as “The prevalence of subclinical hypothyroidism was not…”
Discussion
Line 168: delete “radiation”, it should read “association between internal I-131 exposure and …”
Lines 185-189: provide a reference at the end of the sentence on study by Agate et al.
Line 201: it should read as “than IN those with…” please, correct.
Lines 211-215: The statement describes Agate et al. study findings but has two references. Please correct or revise the sentence.
Line 224: Reference has different format. Please, correct.
Lines 238-239: No logical association between “The same is true for developing nodules” and “as it is well known that age and sex are associated with high prevalence of ATAs and goiter”. Please revise the statement. To reviewer’s opinion, it is more correct to say that female sex and aging are risk factors for ATA-positivity and goiter.
Line 241: please replace for “small” rather than “limited”
Fig 2: the statement following the figure title “No significant difference of diffuse goiter prevalence was observed in both groups 19.7% vs 22.0; p=0.482)” is not related to the data presented on the graph (25.3% vs 26.7%, p=0.780). Please correct the discrepancy and include data on the diffuse goiter prevalence if needed in the Results section.
Table 1: Please, correct for “Characteristics”
Table 1, Footnote: please correct for “shown” without “e”
Table 1, Footnote: please correct for “log-TSH” without capital “L”
Table 3, Head of the 3rd column: Please correct for “goiter” not “goitOr”

Experimental design

Remove a para on study of Shibata et al, 2001 from the Materials and Methods as irrelevant to the study methods. If necessary, please include this information into the
Discussion.

Validity of the findings

Discussion statements (lines 200-298) need substantial revisions, because of authors’ imprecise interpretation of the screening study results in Ukraine and Belarus (Tronko et al., 2006; Ostroumova et al. 2009, 2013). It is strongly recommended to read carefully those publications first to avoid mistakes referring to the same Ukrainian screening cohort as “another screening study” in lines 203-204. In the study of Tronko et al., 2006, no statistically significant association was found between I-131 thyroid dose and prevalence of antibody-positive hypothyroidism and autoimmune thyroiditis (underlined by reviewer) (Tronko et al., 2006). In the same screening cohort statistically significant non-linear relationship between I-131 thyroid dose and ATPO levels was found that was more apparent in euthyroid, thyroid-disease free individuals with moderately elevated ATPO (Tronko et al, 2006). A positive statistically significant association between subclinical hypothyroidism prevalence and I-131 thyroid dose (underlined by reviewer) was clearly established both in Ukrainian and Belarus screening cohorts (Ostroumova et al. 2009, 2013). Assessing ATPO-positivity status as an effect modifier, it was shown that I-131- related risk of subclinical hypothyroidism is higher in study participants with ATPO levels <= 60U/ml compare to those with ATPO levels >60 U/ml.

For the clarity of meaning, the reviewer insists on extensive revisions of the statement “The results of the present study are consistent with the results of these screening studies; the lack of any significantly different prevalence of overt or subclinical hypothyroidism between two groups indicated that there is no significant relationship between I-131 exposure and the prevalence of antibody-positive hypothyroidism…”. It is an overstatement taking into account serious limitations of the study, including a potential for selection bias (only those subjects who visited the examination center), small size of study groups and very few cases of overt and subclinical hypothyroidism (four and nine cases, respectively), and lack of any information on individual levels of thyroid exposure in the study group 1.
It is still missing what authors consider as strengths of their study, what are the novelty and importance of their findings compare to others already published

Additional comments

As a reviewer, I still have some reservations before recommending the paper for publication. Although after the suggested revisions the paper has clearly improved, it still needs revisions to be done. Please see the comments.

---

## Round 0.3 · accepted · Accept

All questions have been answered